**Data Availability Statement:** The raw codes written in MATLAB and Python, a MATLAB GUI, manual, and the non-telecentric holograms are

# Comprehensive tool for a phase compensation reconstruction method in digital holographic microscopy operating in non-telecentric regime

**Brian Bogue-Jimenez** [1], **Carlos Trujillo** [2], **Ana Doblas** [1]*

**1** Department of Electrical and Computer Engineering, The University of Memphis, Memphis, Tennessee, United States of America, **2** School of Applied Sciences and Engineering, Universidad EAFIT, Medellin, Colombia

* adoblas@umassd.edu

## Abstract

Quantitative phase imaging (QPI) via Digital Holographic microscopy (DHM) has been widely applied in material and biological applications. The performance of DHM technologies relies heavily on computational reconstruction methods to provide accurate phase measurements. Among the optical configuration of the imaging system in DHM, imaging systems operating in a non-telecentric regime are the most common ones. Nonetheless, the spherical wavefront introduced by the non-telecentric DHM system must be compensated to provide undistorted phase measurements. The proposed reconstruction approach is based on previous work from Kemper's group. Here, we have reformulated the problem, reducing the number of required parameters needed for reconstructing phase images to the sensor pixel size and source wavelength. The developed computational algorithm can be divided into six main steps. In the first step, the selection of the +1-diffraction order in the hologram spectrum. The interference angle is obtained from the selected +1 order. Secondly, the curvature of the spherical wavefront distorting the sample's phase map is estimated by analyzing the size of the selected +1 order in the hologram's spectrum. The third and fourth steps are the spatial filtering of the +1 order and the compensation of the interference angle. The next step involves the estimation of the center of the spherical wavefront. An optional final optimization step has been included to fine-tune the estimated parameters and provide fully compensated phase images. Because the proper implementation of a framework is critical to achieve successful results, we have explicitly described the steps, including functions and toolboxes, required for reconstructing phase images without distortions. As a result, we have provided open-access codes and a user interface tool with minimum user input to reconstruct holograms recorded in a non-telecentric DHM system.

## Introduction

Digital Holographic Microscopy (DHM) is a quantitative phase imaging modality that relies on optical interferometry to record holograms [1–5], and numerical reconstruction

publicly available on GitHub, https://github.com/OIRL/noteleDHM-Tool.

**Funding:** This research was partially funded by the Vicerrectoría de Ciencia, Tecnología e Innovación from Universidad EAFIT, and National Science Foundation (NSF) grant number 2042563. The funders had no role in library design, implementation and validation, decision to publish, or preparation of the manuscript.

**Competing interests:** The authors have declared that no competing interests exist.

procedures to retrieve amplitude and phase information from micrometric specimens. The phase information encodes the lateral and axial information of the sample, enabling the characterization of both its functional and morphological information. Therefore, DHM has proven to be a powerful metrological tool when non-invasive imaging is desired across various fields and applications [1–5]. In biology, it has been applied to study cells and tissues, offering a powerful tool for developmental biology [6], stem cell research [7], and cancer diagnosis [8]. DHM has also found applications in material science, where it has been used to study the microstructure and mechanical properties of materials such as polymers [9], ceramics [10], and metals [11]. Additionally, DHM has been used for particle analysis in liquids and suspensions, allowing for the determination of particle size, shape, and concentration [12]. DHM has also been applied in microfluidics to study fluid flow and behavior at the microscale, including lab-on-a-chip devices [13]. DHM has also found applications in industrial inspection, where it has been used for non-destructive testing and quality control in industries such as electronics [14] and pharmaceuticals [15]. These applications demonstrate the versatility and potential of DHM as a tool for scientific and industrial applications.

Wavefront aberrations can significantly distort the accuracy of the phase measurements provided by DHM during the hologram recording [16]. The most common distortion is phase aberration due to the tilt angle between the interfering object and reference wavefronts inherent in all off-axis DHM systems. This first-order phase aberration is observable on the recorded holograms as interferential fringes. Correction of this linear phase aberration is known as the phase compensation stage in DHM reconstruction algorithms, in which the shifted object spectrum (e.g., the +1-diffraction term) is centered on the frequency origin. Over the last decade, several research groups have proposed automated compensation methods [17, 18]. These methods reconstruct phase images by minimizing the number of phase wrappings in the final reconstructed phase map. However, a significant limitation of these computational approaches is that they are designed for only telecentric-based DHM imaging systems, assuming that the center of the +1-diffraction term is a maximum value.

Although telecentric-based DHM systems have been validated as intrinsically linear shift-invariant imaging systems [19], the DHM community has not fully adopted this optical configuration. There are four potential reasons hampering its adoption. The first is related to using non-infinity corrected microscope objective lenses, which generate a diverging spherical wavefront in the image space. The second is the need to create compact DHM imaging systems by minimizing the distance between the infinity-corrected microscope objective lens and the tube lens. The third reason is related to the integration of DHM imaging modality with commercial microscopic systems to extend the reach of the DHM technique to a broader community and enable dual-mode fluorescent imaging with QPI [20]. The final reason is the difficulty of ensuring the distance between these lenses to achieve the required afocal configuration. Regardless of the reason, the direct phase reconstruction of holograms captured using non-telecentric DHM systems present a spherical phase distortion, which arises when a converging or diverging spherical wavefront is found on the image space of the DHM system.

Spherical wavefront distortions must be compensated to provide accurate phase measurements across the imaged field of view, converting the non-telecentric DHM system into a linear shift-invariant phase tool. These distortions can arise from several different sources and can be compensated via physical or numerical methods. Among the physical methods, one can record an additional hologram without a sample (e.g., blank hologram) to reconstruct the spherical wavefront experimentally and subtract it from the distorted phase image of the sample under research [21]. The major limitation of this approach is that one should record the blank hologram at least once for each experiment recording. Single-shot physical methods have been proposed to avoid this limitation. They involve inserting an equivalent non-

telecentric imaging system in the reference arm [22] or illuminating the sample with a converging spherical wave whose focus is conjugated with the front focus of the tube lens for full compensation [23].

Despite these physical methods, the spherical wavefront introduced using non-telecentric DHM imaging systems can be compensated computationally. Reported computational approaches have proposed the estimation of the spherical wavefront using Zernike polynomial fitting (ZPF) [24–27], least square surface fitting [27, 28], or principal component analysis (PCA) [29, 30]. The final reconstructed phase image is obtained by multiplying the reconstructed phase image with spherical distortion with the conjugated distribution of the estimated spherical phase mask. The performance of these computational approaches depends significantly on the ability to determine the parameters of the spherical wavefront (e.g., the x- and y- coordinates of its center, and the radius of curvature) with the highest precision. For example, the success of fitting-based implementations [24–28, 31] requires a large area without sample information (e.g., sample-free hologram). Several research groups have proposed using learning-based models to compensate for low and high-order aberrations [32–35], including the spherical wavefront introduced by non-telecentric imaging systems. The method proposed by Nguyen et al. utilizes a Convolutional Neural Network with a U-Net architecture to automatically segment sample information which is then used with ZPF to compensate for remaining aberrations [33]. Similarly, Ma *et al.* propose a two-stage Generative Adversarial Network (GAN) for background segmentation and then use in-painting to create a reference hologram to eliminate background noise from the object information [34]. Although the benefit of these methods is that they can handle both low and high-order aberrations present in the background, their use is limited to sparse samples. Conversely, the resnet-50 model trained by Xin et al. accomplished the task of phase aberration compensation by predicting the coefficients of a standard 2D polynomial which are used to create the conjugated spherical phase map [35]. However, the performance of these learning-based models depends on the training dataset's amount and quality. All the thousands of inputs for the training procedure of the above-mentioned works require paired ground truths, which often must be further supplemented by data augmentation techniques to provide a sufficient dataset.

In 2017, Min et al. proposed a single-shot computational approach to estimate the spherical wavefront of non-telecentric DHM systems based on a spectral analysis of the recorded holograms [36]. This work extends Min's approach by further reducing the number of required parameters needed for reconstructing phase images. The input parameters of our computational method are the sensor's pixel size and the wavelength of the light source. Because the proper implementation of a framework is critical to achieve successful results, the accuracy of the algorithm proposed by Min et al. is highly dependent on the methods used for the thresholding and segmentation of the ±1 diffraction terms and their compact support size. A preliminary computational function to reconstruct non-telecentric holograms was recently implemented in the pyDHM library under the CNT function [37]. This function provides reconstructed phase images without or with minimum phase distortions after finding the best curvature of the spherical wavefront along the two lateral spatial coordinates using two nested for loops, leading to a high processing time. However, in this work, we have explicitly described the steps, including functions and toolboxes, required for reconstructing phase images without distortions. In contrast to Min's method, we have included iterative minimization algorithms [18] as an optional final optimization step to fine-tune the estimated parameters of the spherical wavefront and provide fully compensated phase images. The implementation for each step of the proposed method has been thoughtfully investigated, aiming to develop a generalized computational tool in DHM. Our approach is independent of the sample's size (i.e., not requiring sample-free field of view within the hologram). Additionally,

we validate the performance of this approach for several microscopic samples, including biological and non-biological samples, distorted with a spherical wavefront with different curvature. The proposed method has been implemented in MATLAB 2021a and Python 3.7.1 and is publicly available via GitHub [38], offering an open-source reconstruction tool (i.e., codes and GUI) for the DHM community.

## Off-axis Digital Holographic Microscopy in non-telecentric mode

At their core, DHM systems are simply optical interferometers used to image unstained (e.g., transparent) microscopic samples [1–5]. Fig 1 shows one of many traditional DHM systems, which follows a Mach-Zehnder interferometer configuration. A coherent illumination source (i.e., laser) emits a divergent spherical beam whose focus is conjugated with the front focus of a converging lens (CL), generating a plane wave after the CL lens. A beam splitter (BS1) is then used to split the plane beam into two beams, producing the reference (R) and object (O) wavefronts. The object beam is then reflected by a mirror (M1) to illuminate an object placed at the front focal plane of the microscope objective (MO) lens. The wavefield scattered by the sample is imaged by an imaging system composed of an infinity-corrected MO lens and a tube lens (TL). Since the object is placed at the front focal plane of the MO lens, the image of the sample is always located at the back focal plane of the TL, regardless of the optical configuration between the MO and TL lenses ($z$ distance in Fig 1). The reference plane wave is then reflected directly onto the sensor by a second mirror (M2). The second beam splitter (BS2) combines both the object and reference wavefronts coherently, enabling the recording of their interference pattern (e.g., hologram) onto the sensor of a CCD/CMOS camera located at the back focal plane of the TL (i.e., the image plane of the DHM system). These holograms encode both amplitude and phase information of the complex amplitude distribution scattered by the microscopic sample.

The hologram distribution can be expressed as,

$$h(x, y) = |u_{IP} + r|^2 = |u_{IP}|^2 + |r|^2 + u_{IP}^* \cdot r + u_{IP} \cdot r^*, \tag{1}$$

where $u_{IP}(x,y)$ is the complex amplitude distribution scattered by the sample at the image plane of the TL, and $r(x,y)$ is the complex amplitude distribution of the reference wave. Typically, the reference wave is a uniform plane wave with an intensity equal to one, i.e.,

$$r(x, y) = \exp\left[i\frac{2\pi}{\lambda}\left(x \sin \theta_x + y \sin \theta_y\right)\right], \tag{2}$$

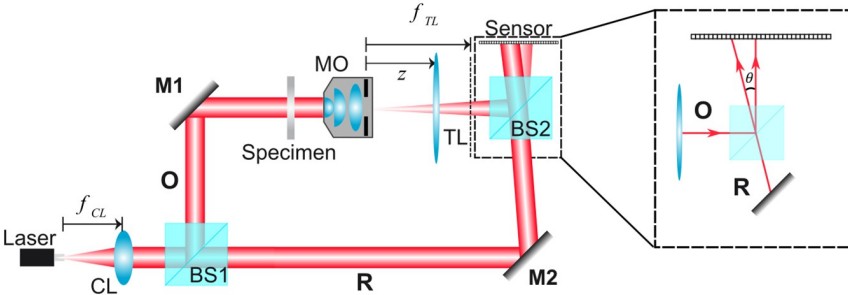

**Fig 1. Traditional Mach-Zehnder DHM system with an imaging system composed of an infinity-corrected microscope objective (MO) lens and a tube lens (TL) operating in non-telecentric regime ($z \neq f_{TL}$).** The sensor records the interference pattern between the object and reference wavefronts. CL, converging lens; BS1 and BS2, beam splitters; M1 and M2, mirrors; O, object wave; R, reference wave.

where $\lambda$ is the wavelength of the coherent light source, and the angle $\boldsymbol{\theta} = (\theta_x, \theta_y)$ is the angle between the wavevectors of the object and reference beams. Assuming an arbitrary object complex amplitude distribution, $o(x, y) = a_0(x, y)\exp[i\, \varphi_0(x, y)]$, the complex amplitude distribution scattered by it and imaged by the imaging system ($u_{IP}$) is equal to

$$u_{IP}(x, y) = \frac{1}{M_L} e^{ikL_0} \exp\left[i\frac{k}{2C}(x^2 + y^2)\right] \left\{o\left(\frac{x}{M_L}, \frac{y}{M_L}\right) \otimes P\left(\frac{x}{\lambda f_{TL}}, \frac{y}{\lambda f_{TL}}\right)\right\} \qquad (3)$$

where $\otimes$ is the 2D convolution operator, $M_L$ is the lateral magnification between the object and image plane, $k = 2\pi/\lambda$ is the wavenumber, $L_0$ is the optical distance between the object and image plane, $C$ is the curvature of the spherical aberration defined by $C = f_{TL}^2/(f_{TL} - z)$ being $f_{TL}$ the focal length of the TL and $z$ the distance between the aperture stop of the MO lens and the TL. If the front focal plane of the TL lens coincides with the plane of the aperture stop (e.g., $f_{TL} = z$), the spherical wavefront in the image space of the DHM system becomes a plane wavefront. Therefore, the spherical phase term in Eq (3) is removed. Overall, the complex wavefield produced by the imaging system is the 2D convolution between a magnified replica of the object distribution and a scaled replica of the Fourier transform of the aperture stop of the MO lens, $P(u,v) = \text{FT}[p(x,y)]$. For the sake of simplicity, in this discussion, we have assumed that the size of the aperture stop is infinite, and therefore we can omit the convolution operation, thus $u_{IP}(x, y) \propto \exp\left[i\frac{k}{2C}(x^2 + y^2)\right] o\left(\frac{x}{M_L}, \frac{y}{M_L}\right)$.

To acquire useful information about the sample, the object information encoded in the hologram [Eq (1)], must be isolated from the object and reference intensities and the virtual image,

$$\begin{aligned} h_F(x, y) &= r^*(x, y) \cdot u_{IP}(x, y) \\ &= a_0\left(\frac{x}{M_L}, \frac{y}{M_L}\right) \exp\left[i\varphi_0\left(\frac{x}{M_L}, \frac{y}{M_L}\right)\right] \exp[it(x, y)] \exp[is(x, y)], \end{aligned} \qquad (4)$$

where $t(x,y)$ represents the argument of the exponential in Eq (2),
$t(x, y) = \frac{2\pi}{\lambda}(x \sin\theta_x + y \sin\theta_y)$, and $s(x,y)$ is the argument of the spherical phase factor of the object wavefield in Eq (3), $s(x, y) = \frac{k}{2C}(x^2 + y^2)$. The amplitude distribution scattered by the sample can be reconstructed by estimating the absolute value of Eq (4), $\hat{a}_0(x, y) = |h_F|$. On the other hand, estimating the object phase from Eq (4) requires the compensation of the interference angle and the spherical wavefront, $\hat{\varphi}_0(x, y) = \text{angle}[h_F] - t - s$.

The interfering angle, $\boldsymbol{\theta} = (\theta_x, \theta_y)$, and the curvature of the spherical wavefront ($C$) can be measured experimentally by analyzing the Fourier transform (FT) of the hologram (e.g., the hologram spectrum). The spectrum of the filtered object is

$$H_F(u, v) = \left[O(M_L u, M_L v) \otimes \delta\left(u - \frac{\sin\theta_x}{\lambda}, v - \frac{\sin\theta_y}{\lambda}\right)\right] \otimes \exp[-i\pi\lambda C(u^2 + v^2)], \qquad (5)$$

where $(u,v)$ are the lateral spatial frequencies, $O(\cdot)$ is the Fourier transform of the complex amplitude distribution. Some irrelevant constant factors have been omitted in Eq (5). The center of the object spectrum is centered at the spatial frequencies $\left(\frac{\sin\theta_x}{\lambda}, \frac{\sin\theta_y}{\lambda}\right)$. Assuming that the hologram is recorded onto the surface of a discrete sensor with $X \times Y$ square pixels of $\Delta_{xy}$ size, the interference angle can be calculated as [18]

$$\theta_x = \sin^{-1}\left(\frac{|u_0 - P|\lambda}{X\Delta_{xy}}\right), \qquad (6)$$

and

$$\theta_y = \sin^{-1}\left(\frac{|v_0 - Q|\lambda}{Y\Delta_{xy}}\right),\tag{7}$$

where $(u_0, v_0)$ are the lateral pixel position of the center of the DC term, being equal to $u_0 = (X/2)+1$ and $v_0 = (Y/2+1)$, and $(Q, P)$ are the center pixel positions of the +1 term in the Fourier domain, $H_F(u,v)$.

In Eq (5), the spectrum of the object is convolved by a spherical wavefront in the Fourier domain. This spherical wavefront makes the compact support of the +1-term change from a circle in telecentric-based DHM systems to a rectangle in non-telecentric DHM systems [39]. The size of the +1 term (e.g., $M$ and $N$ along the horizontal and vertical direction, see Fig 2) changes linearly with the inverse of the curvature of the spherical wavefront (1/$C$) [39]. In fact, Sanchez-Ortiga *et al.* demonstrated that the lateral size of the +1 term is equal to $M = (X\Delta_{xy})^2 / (\lambda C)$ along the x coordinate and $N = (Y\Delta_{xy})^2 / (\lambda C)$ along the y coordinate. Therefore, the size of the compact support of this term in each direction, or simply, the curvature in the spatial domain, can be estimated as

$$C_x = \frac{\left(X\Delta_{xy}\right)^2}{\lambda M},\tag{8}$$

and

$$C_y = \frac{\left(Y\Delta_{xy}\right)^2}{\lambda N}.\tag{9}$$

Eqs (8) and (9) show that the curvature of the spherical wavefronts in Eqs (3) and (5) may differ between the two lateral coordinates if the hologram is not square ($X \neq Y$).

## Reconstruction of quantitative phase images in non-telecentric DHM systems

This section describes how the proposed reconstruction algorithm for quantitative phase imaging using non-telecentric DHM systems has been implemented, Fig 2. As previously mentioned, off-axis DMH systems operating in non-telecentric mode are popular even though they result in holograms with spherical phase distortions. Due to the popularity of such systems, many variations of these phase distortions are possible based on the experimental configuration of the DHM systems. Our experimental holograms have different experimental conditions, including MO lenses with different lateral magnification ($M_L$) and numerical aperture (i.e., spatial resolution). In particular, we used two infinity-corrected Nikon microscope objective lenses: a 20×/0.5 NA and 40×/0.75 NA microscope objective lens. Also, the experimental holograms differ in the interference angle between the object and reference beams. The distance between the aperture stop of the MO lens and the TL also changes, providing spherical wavefronts with different curvatures. In all the experimental holograms, the wavelength of the illumination laser was $\lambda$ = 532 nm, and the camera used to record the holograms had a pixel size of $\Delta_{xy}$ = 5.86 μm. We have used experimental holograms of the Benchmark Technologies Quantitative Phase Target (QPT™) and a smearing sample of human red blood cells from Carolina Biological Supply Company (item # C25222) to validate the proposed methodology. The steps of the proposed algorithm (Fig 2) are based on Min's approach [37]. The

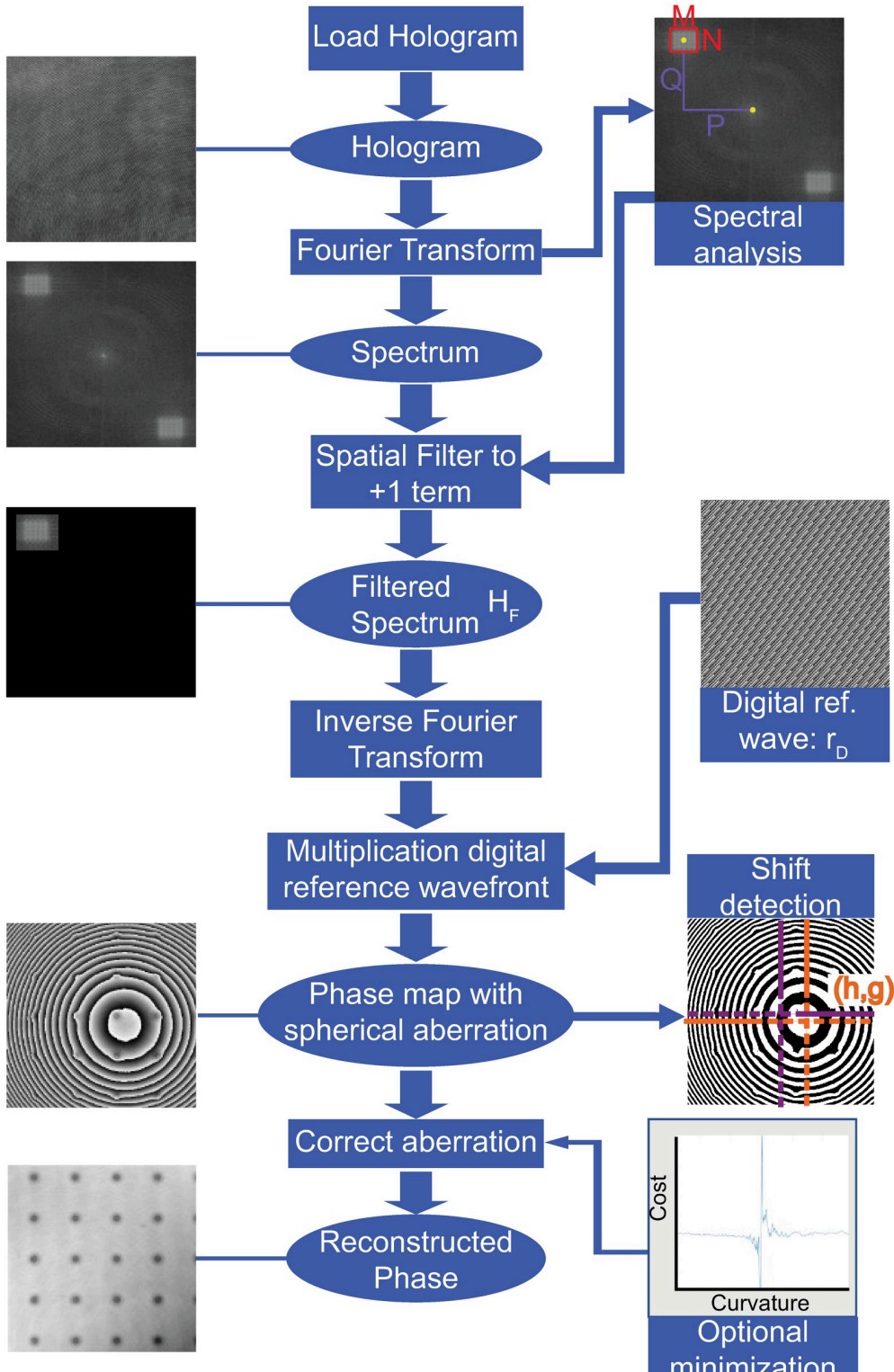

**Fig 2. Flowchart outlining the spectral analysis reconstruction algorithm for quantitative phase imaging using non-telecentric DHM systems.**

implementation has been performed in MATLAB using a 2021a academic license and its image-processing toolbox, and in Python 3.7 using the libraries scikit-image and scipy. A guided user-interface (GUI) has also been developed in both software platforms for ease of use. After loading the hologram, the fast 2D Fourier transform (FFT) is computed. It is important to reorganize the hologram spectrum to shift the zero-frequency component to the center of the hologram spectrum using the widely known FFT shifting function. The next step is related to the identification of the +1 term [Eq (5)]. Automatic approaches for image segmentation were considered to identify the +1 term. However, the accuracy of these methods was highly variable, often producing under- or over-estimates of the +1 term. Therefore, for this step, we rely on the user to identify the desired crop rectangle in the spectral domain of the hologram. This rectangle will define the initial values of the +1 term's position ($P$ and $Q$) and shape ($M$ and $N$). The user ought to slightly overestimate the rectangles' size, to ensure higher frequency information is not lost, which can later be fine-tuned by using the sliders (shown in Fig 3). After the hologram's ±1 term has been identified, the interference angle of the DHM system [Eqs (6) and (7)] can be defined by the position of the +1 term's centroid.

The interference angle, $\boldsymbol{\theta} = (\theta_x, \theta_y)$, is given by the distance from the +1 term's centroid to the spectrum's center, see Eqs (6) and (7). The next step is related to the compensation of the interference angle between the reference and object waves. For this step, we must multiply the inverse Fourier transform of the filtered hologram's spectrum [$h_F$, Eq (4)] by a discrete digital reference wave $r_D(m,n)$

$$r_D(m, n) = \sum_{m,n}^{X,Y} \exp\left[ i\frac{2\pi}{\lambda} \left( m X \sin\theta_x + n Y \sin\theta_y \right) \Delta_{xy} \right], \tag{10}$$

where ($m,n$) are the discrete lateral coordinates of the sensor, $X$ and $Y$ are the number of pixels

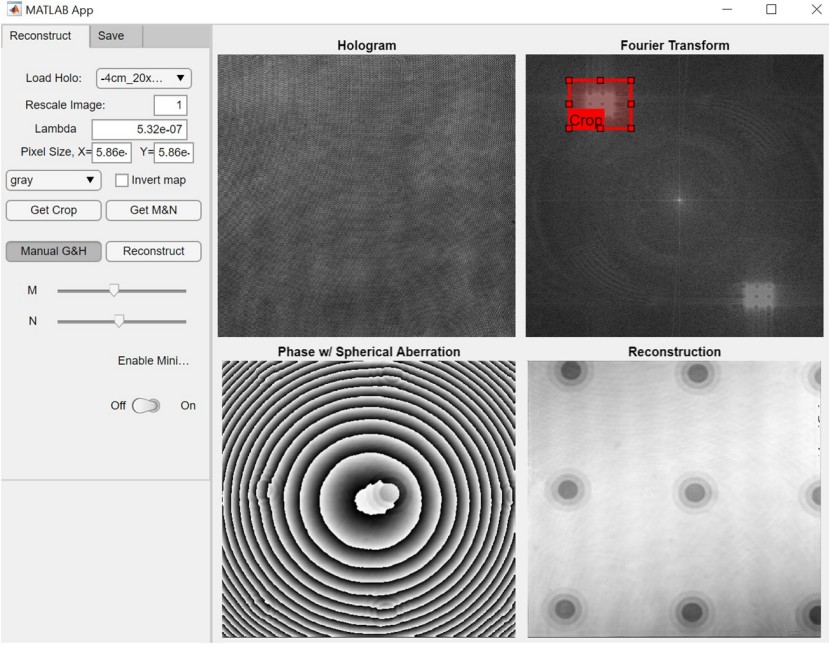

**Fig 3. Identification of the ±1 term from the hologram's spectrum using the proposed MATLAB GUI.** The curvature of the spherical phase factor can be estimated from the height (M) and width (N) of the ±1 terms, identified by the user, using Eqs (8) and (9). The interference angle can be estimated using the distance between the center of the ±1 term and the spectrum's origin (via P and Q) using Eqs (6) and (7).

of the sensor in each direction, and $\Delta_{xy}$ are the lateral dimension of its square pixels. The computation of the digital reference wave in Eq (10) requires the interference angle, which is given by Eqs (6) and (7) (and an example is shown in Fig 4).

Next the curvature of the spherical wavefront introduced by the non-telecentric configuration [Eqs (8) and (9)] can be estimated from the shape of the ±1 diffraction terms in the Fourier spectrum, see Fig 3. The right panel in Fig 4 shows the reconstructed phase image after compensating the interference angle in non-telecentric DHM systems. The partially compensated phase image shows a ring-like pattern superimposed over the sample's phase distribution (Figs 4 and 5a). This ring-like pattern is directly related to using a non-telecentric imaging system since the object distribution is distorted by a spherical phase factor, as Eqs (3) and (4) show.

The spherical wavefront distorting the phase distribution, $s(x,y)$, must be compensated to provide accurate phase measurements. One can generate a conjugated digital spherical wavefront knowing its curvature ($C_x$, $C_y$) and center ($h$, $g$) as

$$u_S^*(m, n) = \sum_{m,n}^{X,Y} \exp\left[-i\frac{\pi}{\lambda}\left(\frac{(m-h)^2}{C_x} + \frac{(y-g)^2}{C_y}\right)\Delta_{xy}^2\right] \tag{11}$$

The curvature of the spherical wavefront along the $x$- and $y$- directions (e.g., $C_x$ and $C_y$) are given by the size of the +1 term, as Eqs (8) and (9). The center of the distorted spherical wavefront ($h$, $g$) can be estimated from the reconstructed phase map (right panel in Fig 5) by binarizing that image using Otsu's global thresholding. We have used the *regionprops* function with the lowest "Eccentricity" property to ensure that the bounding box completely encompasses an entire ring. Fig 5 shows an example of the estimation of the center of the spherical wavefront.

Fig 6(a) shows the reconstructed 2D phase map of a star target from the Benchmark Technologies Quantitative Phase Target (QPT™). Panel (b) in Fig 6 shows the radial phase profile at two different radii: $r = 43.95$ μm (pink), and $r = 73.25$ μm (cyan). We have also plotted the nominal phase value, marked the gray-shaded area in Fig 6(b). The nominal phase values have been calculated based on the manufacturer's specifications, a refractive index and thickness equal to 1.52 and 350 nm, respectively. There is a high similarity between the experimental phase values and the nominal ones, demonstrating the accuracy of the proposed method to compensate spherical phase distortions. The results illustrated in Fig 6 confirm that, within

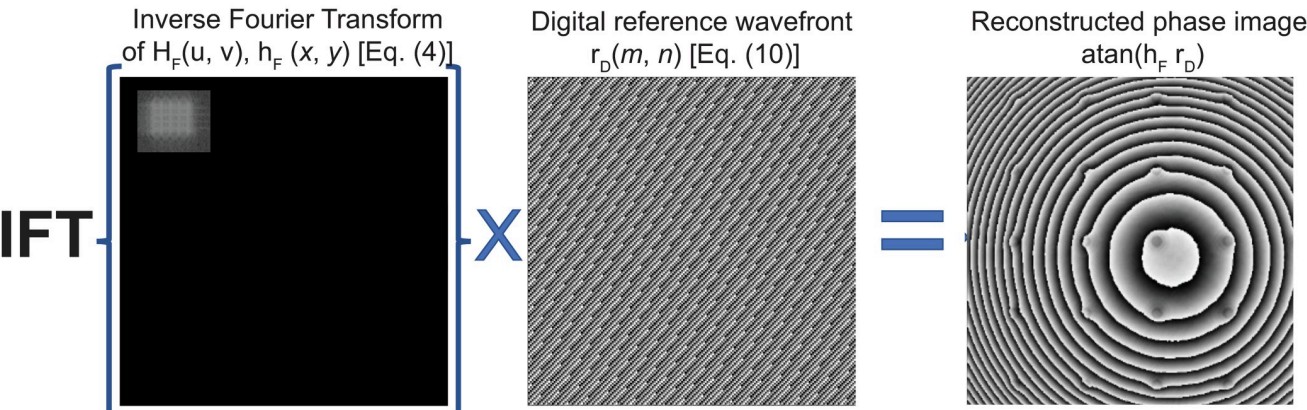

**Fig 4. Compensation of the interference angle by multiplying the inverse Fourier transform of the filtered hologram's spectrum with a digital reference wave.** A spherical wavefront related to using a non-telecentric imaging system still distorts the reconstructed raw phase image.

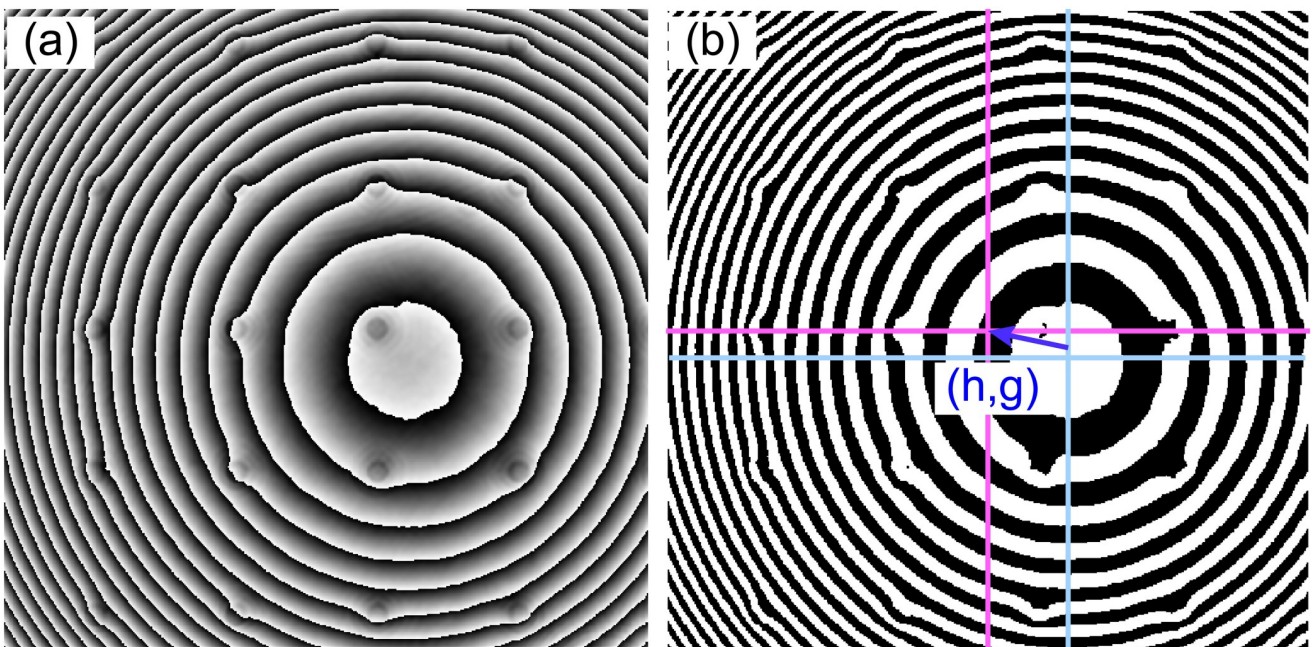

**Fig 5. Performance of the automated step to find the center (*g* and *h*) of the spherical wavefront by binarizing the reconstructed raw phase image.** Panel (a) shows the reconstructed phase image after compensation by the linear phase related to the off-axis configuration. Panel (b) shows the binarized phase with the true center of the image (magenta) and the center of the spherical aberration (cyan).

experimental errors, the spherical phase factor introduced by the non-telecentric configuration has been compensated, providing a linear shift-invariant quantitative phase imaging tool.

For further quantification of the proposed method, Fig 7(a) and 7(c) show the reconstructed 2D phase images of a USAF target and wedding cakes. This second experiment aims to validate the performance of our method to reconstruct uniform background values, proving that the distorting spherical wavefront has been effectively removed. We have plotted some vertical and horizontal background phase profiles in Fig 7(b) and 7(d) measured at the colored lines over the phase images in Fig 7(a) and 7(c). The comparison of these background profiles in Fig 7(b) and 7(d) confirms that the low frequency phase information is fairly uniform across the different directions Aside from minimal discrepancies from a complete flat background, these results demonstrate the effectiveness of our proposed method to compensate spherical distortions in all directions. Nonetheless, some profiles in these panels show the presence of a residual spherical wavefront. In particular, this residual spherical wavefront is clearly identified on the profiles of the wedding cakes (Fig 7d).

Although this residual spherical wavefront can be removed by manually adjusting M and N values, users must devote some time to reconstructing a fully compensated phase map of the sample without any linear or spherical aberrations. We understand that this manual compensation can be arduous for inexperienced users. Therefore, alternatively, we have included a final optional step to reconstruct phase distributions without phase aberrations based on minimizing a cost function. Two cost functions have been identified within the DHM community. The first one is based on the prior work of Trujillo et al in 2016, who demonstrate that the binary phase image from the best compensated phase map should be all white [17]. Based on this observation, in 2021, Castaneda et al. proposed a cost function ($J_1$) that counts the total number of phase wraps in the binary reconstructed phase image [18]. Alternatively, other researchers have proposed an automatic phase aberration compensation method based on

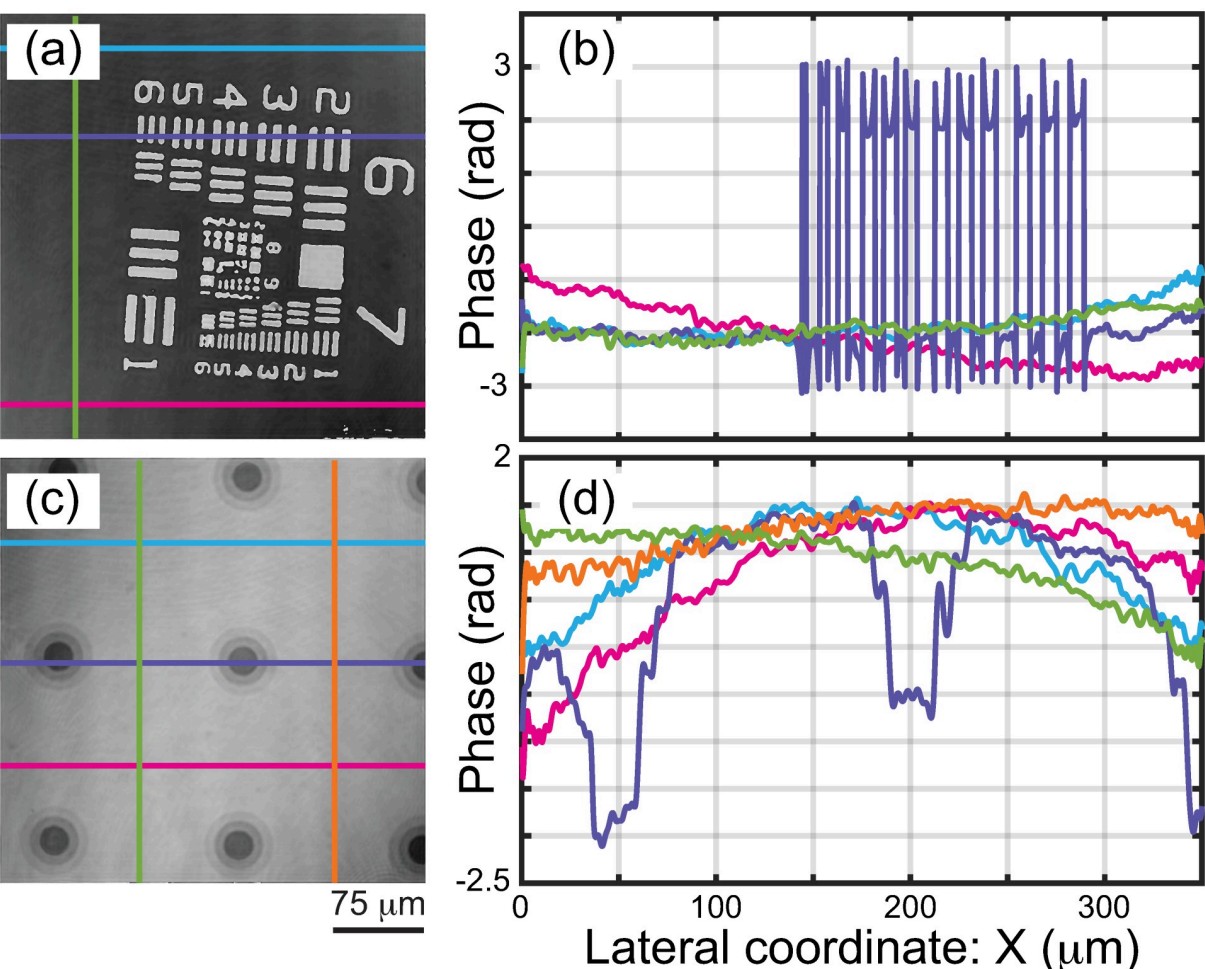

**Fig 6.** (a) Reconstructed phase images of a star target from the Benchmark Technologies Quantitative Phase Target (QPT™). (b) Radial phase profiles at radii of $r$ = 43.95 μm and $r$ = 73.25 μm over the 2D phase map, shown as pink and cyan circles in panel (a), respectively.

phase variation minimization [40, 41]. In other words, the second cost function ($J_2$) measures the reconstructed phase map's standard deviation (SD). Our preliminary comparison between both cost functions (not shown here) shows that the SD-based cost function produces a more uniform distribution of the background phase values. Nonetheless, we have implemented both cost functions in the proposed computational approach to allow each user their use.

Finally, we have compared the performance of the proposed method with the one provided by the subtraction method, which uses a blank hologram to compensate for the spherical aberrations caused by a non-telecentric alignment [21] via the direct subtraction between both reconstructed phase maps. Fig 8 shows the normalized reconstructed phase images of a USAF phase target for the proposed method (Fig 8a) and the subtraction one (Fig 8b), demonstrating the high agreement between both methods. The accuracy and resolution of both methods have been evaluated by plotting the normalized phase values along the vertical direction [indicated by the white arrows in Fig 8(a) and 8(b)] of the 9 group, see Fig 8(c). From these profiles, one can identify the minimum resolvable element of the USAF phase target is the 9–3 element, which corresponds to 0.78 μm. This value confirms that both methods provide reconstructed images operating at the system's coherent diffraction limit, d = λ/NA = 0.532 / 0.75 = 0.71 μm.

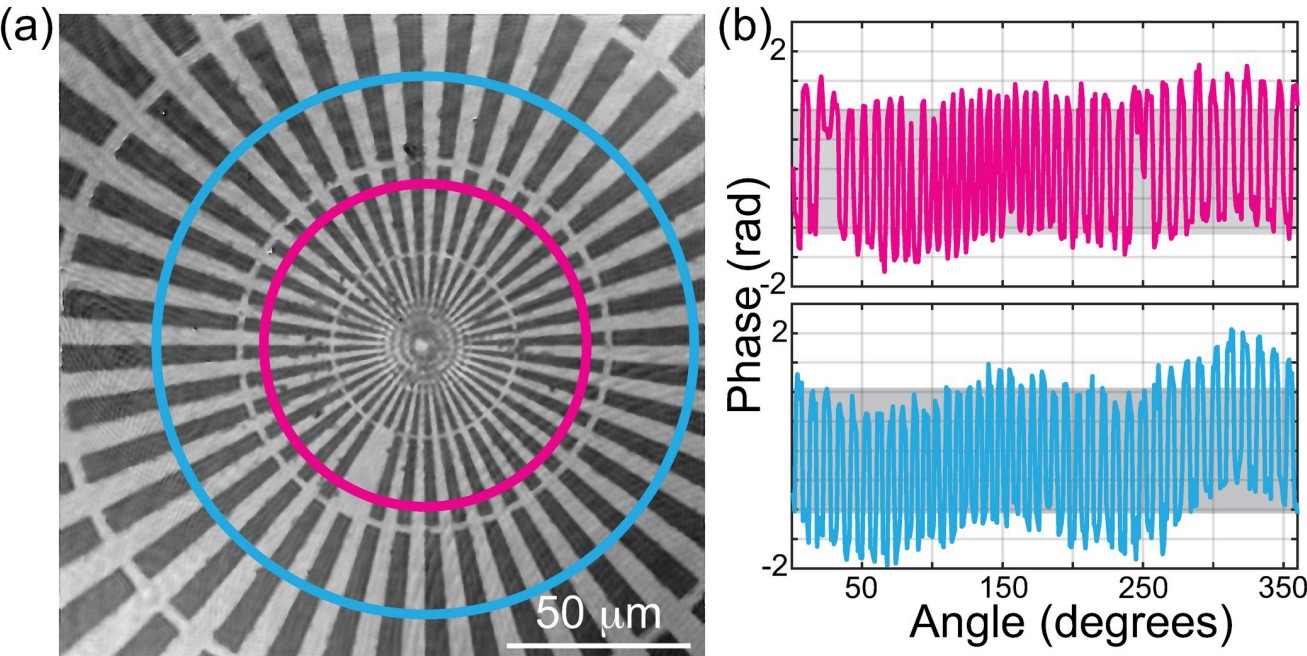

**Fig 7.** (a, c) Reconstructed 2D phase images of a USAF target and wedding cakes. (c, d) Vertical and horizontal phase profiles along the colored lines over the corresponding phase images.

Despite that the dual-shot subtraction method is simple to perform and computationally inexpensive, its effectiveness is highly dependent on the experimental conditions during data acquisition, requiring that the blank hologram must represent an exact replica of the spherical distortions to produce accurate phase distributions. If the experimental DHM system suffers from external factors such as vibrations and temperature fluctuations, the blank and sample holograms do not have the exact same distortion and the subtraction method will not produce accurate results, requiring the use of additional computational methods to reduce any residual distortion. This negative result is further exacerbated during the acquisition of large datasets, where the temporal and spatial changes are most likely to occur. However, the proposed method alleviates such constraints by taking advantage of the single shot nature of off-axis DHM

Among the different minimization algorithms, we have implemented 7 different minimization algorithms to be used with either of the two cost functions in both the codes and GUIs. The minimization algorithms are: FMC, FMU, FSO, SA, PTS, GA, and PS. A short description of these algorithms is found in Section S1 Appendix. All these minimization algorithms are included in the Optimization and Global Optimization MATLAB toolboxes and the *scipy* library from Python. We have also implemented a hybrid optimization option combining the GA and PS methods. In the hybrid optimization approach, firstly, the GA algorithm runs, and after the GA method reaches convergence, a PS algorithm starts a fine-tuning search for the best parameters. The combination of these two techniques was chosen as they individually performed well, and the hybrid GA+PS optimization yields the best results. Initial points on these minimization algorithms are the values found through the manual process of the spherical wavefront compensation. No equality or inequality constraints were enforced. The lower and upper bounds were defined with a ±50% range around the initial points for the curvature. Therefore, if the seeded value were 1, the algorithm is given the range from 0.5 to 1.5 to

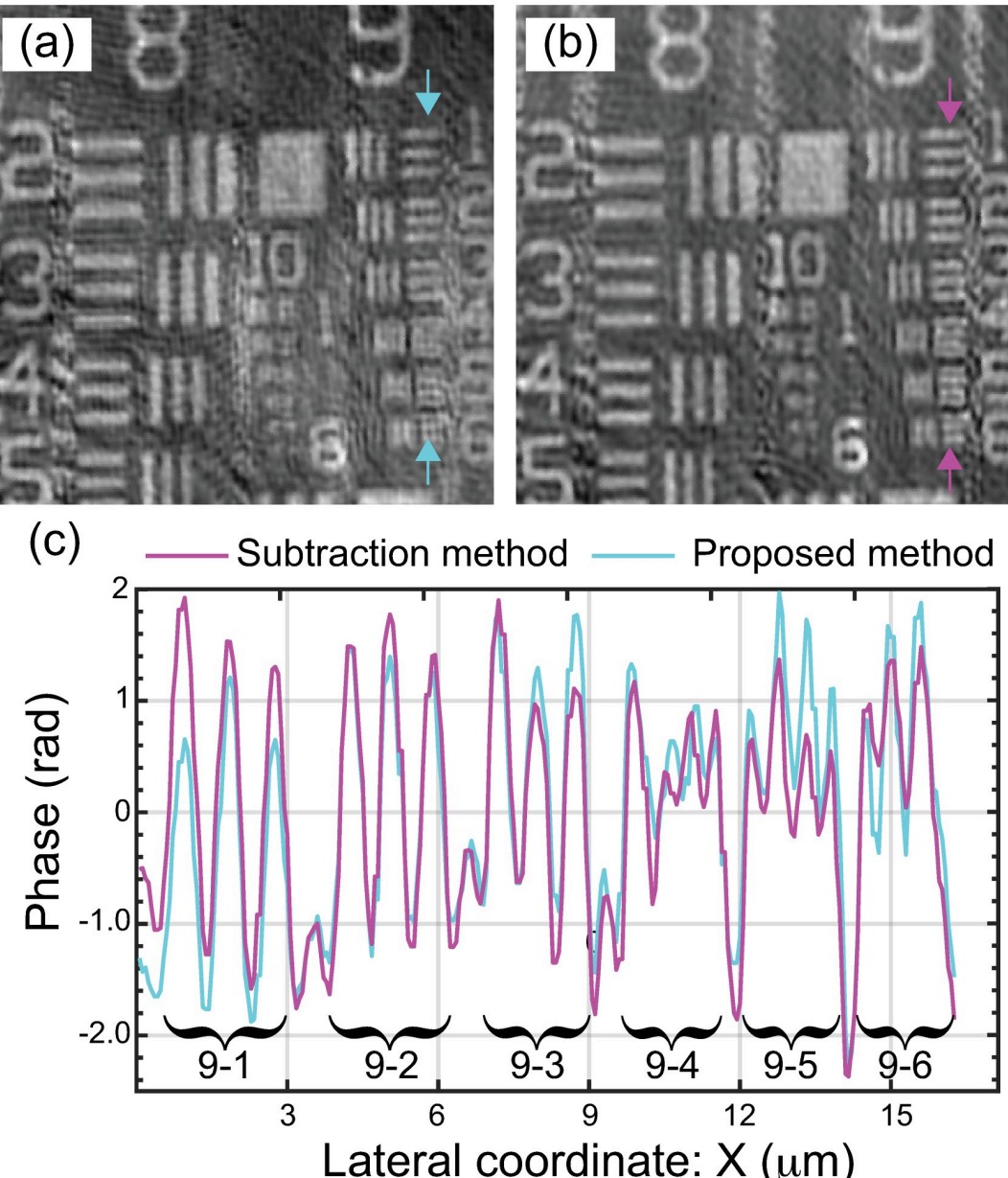

**Fig 8.** Reconstructed phase images of a USAF phase target using the (a) proposed method and (b) subtraction method. (c) Phase profile along the vertical direction (marked by the color arrows in panels a and b) through the horizontal lines of group 9.

explore. The population size for the GA algorithm within the hybrid GA + PS algorithm was limited to 15; this value was found experimentally to improve convergence speed without sacrificing accuracy. All other values were left as the default since they were not found to significantly improve the speed or accuracy in the final reconstructed phase images. The comparison between the different minimization algorithms is Table B1 in S2 Appendix. Two metrics are used to evaluate the performance of these algorithms using the reconstructed phase images. These metrics are the Percent Error (PE), the Structured Similarity Index Measure (SSIM),

and the average time. The processing time is reported based on a Windows-based i7-8700 K CPU (3.70 GHz) 16.0 Gbyte RAM desktop computer. The PE compares the estimated curvature of the spherical wavefront along the two lateral directions ($C_x$, and $C_y$) obtained after applying the minimization algorithm and the manual estimation of those parameters using an iterative looping. For example, if the estimated $C_x$ value using the GA minimization algorithm is equal to 0.5769, and the manual estimation of $C_x$ is 0.5707, the PE value is equal to 1.08% (e.g., PE = |(0.5769–0.5707)|/ 0.5707 × 100%). The SSIM metric compares the reconstructed phase image for each algorithm and the ground truth phase map obtained after the intensive iterative looping. For a complete statistical analysis, Table 1B in S2 Appendix reports the mean and the standard deviation values of the PE and SSIM metrics as well as the range of the PE metric within the tested experimental holograms. The results in Table 1B in S2 Appendix shows that the combination of the GA and PS minimization algorithms provides the highest similarity between the reconstructed phase images obtained after the minimization process and the ground truth phase image with an average SSIM value of 0.634 and a standard deviation (std) of 0.37. On average, the hybrid GA+PS approach takes approximately a minute per image to find the $C_x$ and $C_y$ values and reconstruct phase images with minimum phase distortions. Although the fastest minimization algorithm is the FMU function with an average processing time of 2.96 seconds, the reconstructed phase images present significant phase distortions, leading to an almost null SSIM value. The PS algorithm is also quite fast, taking about 4 seconds on average to find the minimum values. Nonetheless, the similarity between the reconstructed phase images is slightly reduced from 0.634 with the GA+PS approach to 0.555 with the PS method.

Fig 9 shows the performance of the hybrid GA + PS algorithm to compensate for the spherical wavefront. Panel (a) shows the reconstructed phase image with a residual spherical wavefront due to an improper estimation of the curvature of the spherical wavefront. Panel (b) provides the final reconstructed phase image after minimizing the SD-based cost function using the GA + PS minimization algorithm. One can realize that no spherical wavefront distorts the reconstructed phase map in Fig 9(b). In addition, the background of the phase distribution in Fig 9(b) is uniform, confirming the correct compensation of any linear and spherical aberration. To better visualize the background uniformity, we have unwrapped the reconstructed 2D phase image [Fig 9(c)] and show the three-dimensional pseudo color phase image in Fig 9(d). We have used the unwrapping method described in Ref. [42]. According to the color bar in Fig 9(d), the difference between the phase values in the background is negligible, confirming the success in computationally removing any residual term. Also, the phase values of the wedding cakes placed at different regions within the field of view are the same within experimental errors (see the cross-sectional profile in Fig 9(e)), proving that the proposed method provides shift-invariant phase measurements.

Finally, we have tested the proposed computational approach's performance by reconstructing a 2D phase image from red blood cells (RBCs). Fig 10 shows the reconstructed 2D and 3D phase map of an RBC sample. Again, neither linear nor spherical aberrations are present in Fig 10, confirming the success of the proposed tool in compensating any phase term related to the off-axis non-telecentric DHM system.

## Conclusions

In conclusion, this work comprehensively describes the reconstruction and phase compensation of holograms recorded using an off-axis DHM system operating in a non-telecentric regime. This work offers a step-by-step process for implementing a computational method that compensates for both tilt and spherical aberrations using spectral analysis. This approach's

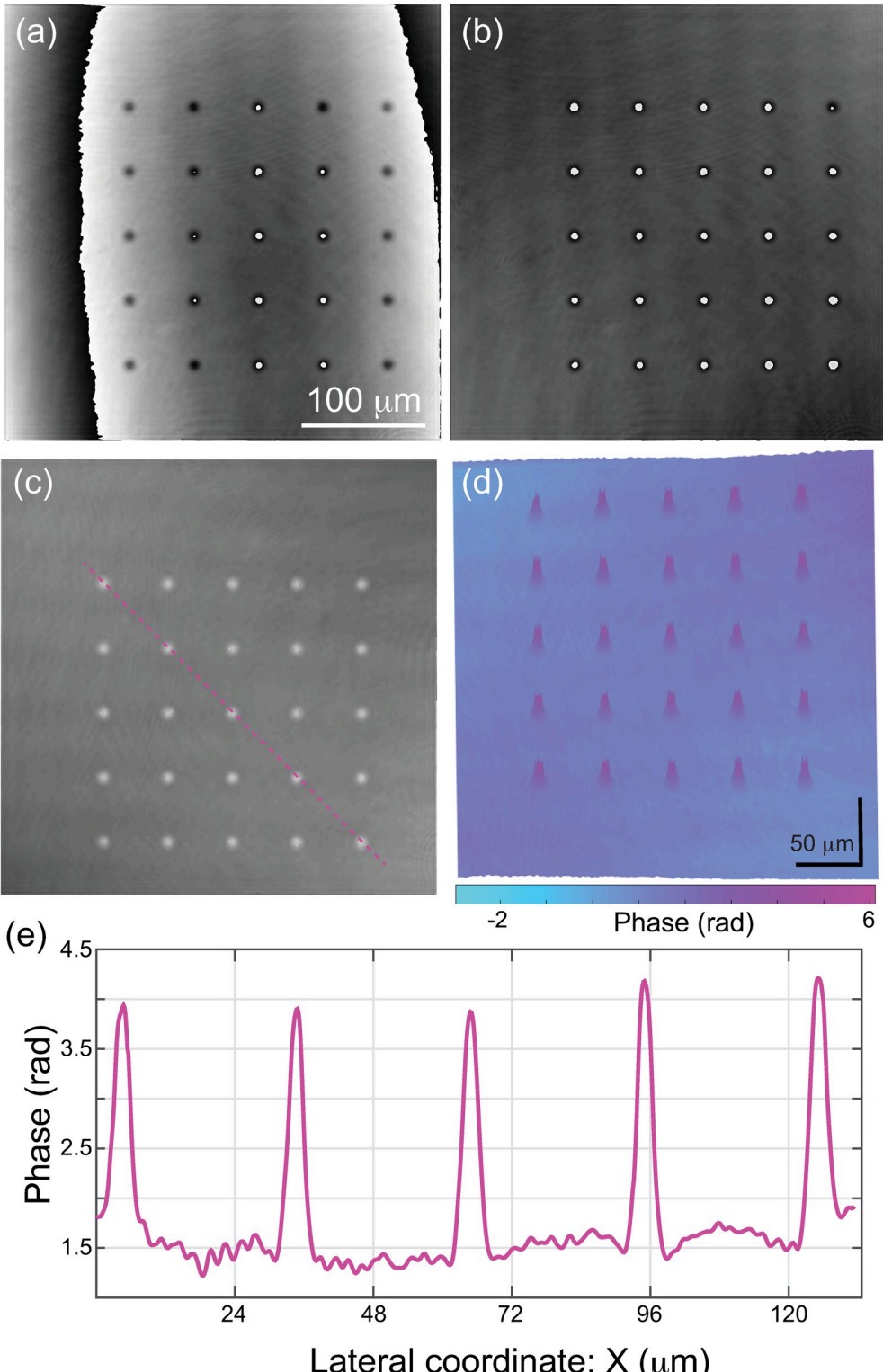

**Fig 9.** (a, b) Reconstructed phase images of wedding cakes before (a) and after (b) applying a minimization algorithm to computationally remove any distorting linear and spherical phase aberrations. (c) Unwrapped reconstructed phase image. (d) Three-dimensional pseudocolor image of the phase map shown in panel (c). (e) Cross-sectional profile view of phase along the pink direction in panel (c).

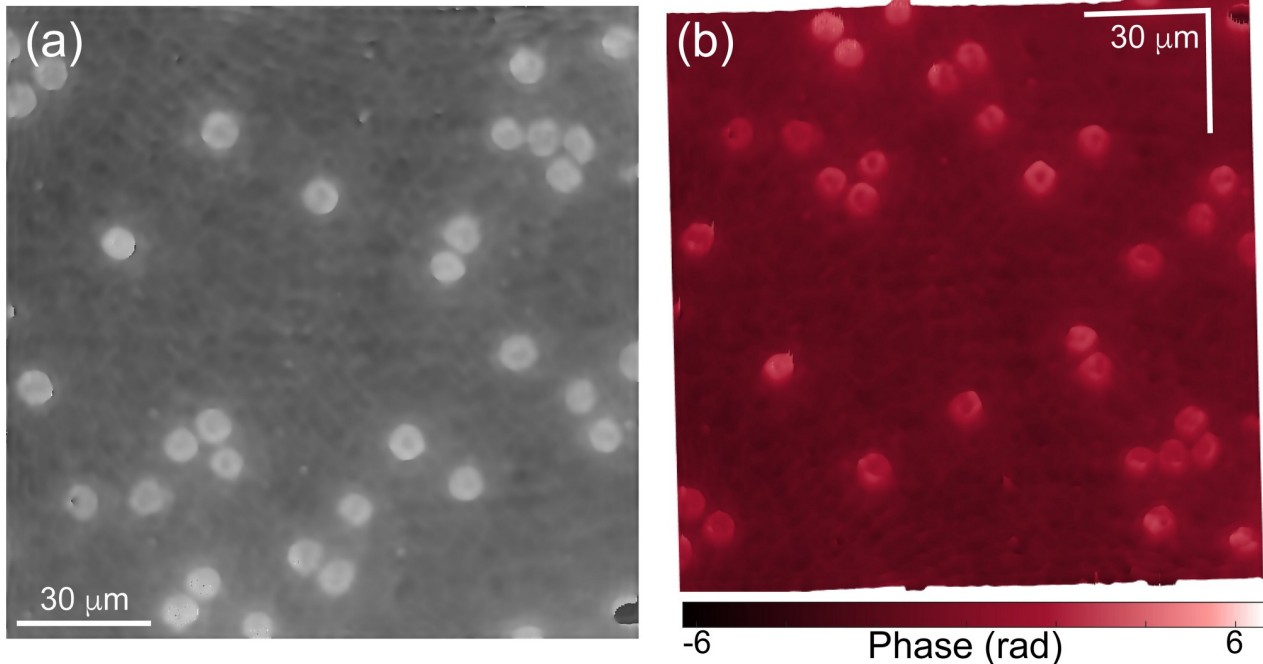

**Fig 10. Reconstructed 2D (a) and 3D (b) phase image of an RBC sample using a non-telecentric DHM system.**

source code is written in MATLAB 2021a and Python 3.7.1 and is publicly available via GitHub. To increase the applicability of the proposed method, we also provide some instructional videos on how to use our tool [43]. Our implementation offers research and educational tools that benefit the DHM community. The most obvious of these benefits is the reconstruction of holograms by users with minimal knowledge of the system used to capture the images. Although the proposed computational approach has been validated with non-telecentric DHM imaging systems and reference plane waves, it can be used for any off-axis DHM system in which a spherical wavefront distorts the complex object distribution in the recorded hologram. This distorting spherical wavefront can come from the object illumination, the reference illumination, the imaging system or all the above. Furthermore, this tool is ideal for creating labeled datasets that can be used to train machine learning or artificial intelligence algorithms for the same purpose. Finally, as an educational tool, this work offers an easy way for the next generation of researchers to understand how these spherical aberrations affect holograms and what steps are needed for their compensation. Future work should explore automated thresholding and segmentation procedures to automatically spatial filter the hologram spectrum and identify the parameters obtained through spectral analysis. This would allow for even less user intervention in getting accurate reconstructed phase images and hopefully improved reconstruction at higher speeds.

## Supporting information

**S1 File.**
(DOCX)

**S1 Appendix. Minimizing algorithms.**
(DOCX)

**S2 Appendix. Comparison between the different minimization algorithms.** (DOCX)

## Author Contributions

**Conceptualization:** Brian Bogue-Jimenez, Carlos Trujillo, Ana Doblas.

**Funding acquisition:** Carlos Trujillo, Ana Doblas.

**Investigation:** Carlos Trujillo, Ana Doblas.

**Methodology:** Brian Bogue-Jimenez, Carlos Trujillo, Ana Doblas.

**Software:** Brian Bogue-Jimenez, Carlos Trujillo, Ana Doblas.

**Supervision:** Carlos Trujillo, Ana Doblas.

**Validation:** Brian Bogue-Jimenez, Carlos Trujillo, Ana Doblas.

**Writing – original draft:** Brian Bogue-Jimenez, Carlos Trujillo, Ana Doblas.

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
