## [Decision Letter · Decision Letter 0]

6 Jun 2023

PONE-D-23-14581

Comprehensive Tool for a Phase Compensation Reconstruction Method in Digital Holographic Microscopy Operating in Non-Telecentric Regime

PLOS ONE

Dear Dr. DOBLAS,

Thank you for submitting your manuscript to PLOS ONE. After careful consideration, we feel that it has merit but does not fully meet PLOS ONE’s publication criteria as it currently stands. Therefore, we invite you to submit a revised version of the manuscript that addresses the points raised during the review process.

We look forward to receiving your revised manuscript.

Kind regards,

Ireneusz Grulkowski, PhD

Academic Editor

PLOS ONE

Journal Requirements:

   "Funding. This research was partially funded by C. Trujillo’s and A. Doblas’ funding resources. C. Trujillo acknowledges the support provided by Vicerrectoría de Ciencia, Tecnología e Innovación from Universidad EAFIT. A. Doblas acknowledges the support provided by National Science Foundation (NSF) through her NSF CAREER grant (grant number 2042563)."

Reviewers' comments:

Reviewer's Responses to Questions

**Comments to the Author**

1. Is the manuscript technically sound, and do the data support the conclusions?

Reviewer #1: Partly

Reviewer #2: Yes

2. Has the statistical analysis been performed appropriately and rigorously? 

Reviewer #1: N/A

Reviewer #2: N/A

3. Have the authors made all data underlying the findings in their manuscript fully available?

Reviewer #1: Yes

Reviewer #2: Yes

4. Is the manuscript presented in an intelligible fashion and written in standard English?

Reviewer #1: Yes

Reviewer #2: Yes

5. Review Comments to the Author

Reviewer #1: In the submitted manuscript, the authors present a computational method to process the phase images acquired with a non-telecentric, digital-holographic microscope. The proposed computational method is thoroughly described. Here are some suggestions.

In digital holographic microscopy, the phase modulation not induced by the sample can be easily and accurately subtracted using a background image, which is acquired for an empty field of view. The same background image can be used repeatedly for multiple sample images acquired afterward; thus, the process does not increase the data acquisition time much. The computational method proposed by the authors would be useful if it provides the same resolution and phase accuracy as the background subtraction method. That said, I’d strongly recommend the authors to perform a more thorough characterization of the proposed method.

- The resolution measurement has not been performed. This could be done by analyzing the edge of the star-target image (Figure 6).

- More important, the phase measurement accuracy needs to be shown using samples of known refractive index values (e.g., polymer microspheres in index-matching liquid).

- The low-frequency phase distribution in the background region (i.e., non-sample region) would be useful to confirm the effectiveness of the proposed method.

Reviewer #2: The authors report about a method for compensation of phase aberrations in quantitative phase imaging (QPI) with non-telecentric off-axis digital holographic microscopy (DHM). After an explanation of the underlying principles and characterization of the method by utilizing a phase test chart the application on a technical sample and red blood cells is illustrated.

In general, the manuscript is motivated, organized, an includes adequate references. The experimental investigations appear to be accurately performed. The results are plausible. The authors address an important topic in QPI with DHM: The compensation of spherical phase aberrations which may be of interest for the field of DHM and the interdisciplinary areas of QPI and label-free biomedical imaging. In summary, the content of the manuscript appears to be suitable for the journal PLOS one.

However, the authors should consider revisions:

1. Abstract: From the abstract the novelty aspects of the proposed phase compensation concept with respect to reference 35 becomes not fully clear. The authors should clarify the abstract concerning this topic.

2. Introduction:

a. To complete the description of the-state-of-the-art the authors may consider adding that non-telecentric arrangements can simplify the combination of DHM with commercial optical microscopes as, for example, reported in Drug Deliv. and Transl. Res. 12, 2207–2224 (2022).

b. In the last paragraph of the introduction the novelty aspects/extensions of the proposed phase compensation concept with respect to reference 35 become not fully clear. The authors should consider adding further clarifying details.

3. Section 2 “Off-axis Digital Holographic Microscopy operation in non-telecentric mode”:

a. The authors may consider removing “operation” from the section title.

b. In general, the explanations in this section include many details and are partly difficult to understand. The author may consider shortening and clarifying the text and to further emphasize the most important statements/topics.

c. Fig. 1: The authors show the sketch of a specific experimental arrangement in which sample illumination and reference wave are plane waves. However, in practice, e.g., a sample illumination via a condenser lens (for illustration see, e.g., ref. 35 of the manuscript or Drug Deliv. and Transl. Res. 12, 2207–2224 (2022)) or utilization of a spherical reference wave can also result in a spatial frequency spectrum as illustrated in Fig. 2 of the manuscript. The authors thus may consider adding a discussion concerning the possible transfer of their approach to a more general regime.

4. Section 3:

a. Figs. 3 and 5 (major point): From the explanations it becomes not fully clear how the spherical phase aberrations are compensated and what are the differences in the procedure reported in reference 35 of the manuscript. For, example: Is equation 10 multiplied or subtracted from phase images like shown in Fig. 5a? The authors should add substantial clarifying information concerning this topic (and perhaps may extend Fig. 4 for an additional illustrating/clarifying sub figure?).

b. Figs. 7 and 8: The investigated sample “wedding cakes” should be explained with more details. The authors may consider adding cross-section plots through the phase images in Figs. 8c and 9a.

c. 2nd paragraph below Fig. 7: The description concerning the minimization algorithms should be supported by data, perhaps, in a figure or table.

6. PLOS authors have the option to publish the peer review history of their article (what does this mean?). If published, this will include your full peer review and any attached files.

Reviewer #1: No

Reviewer #2: No

---

## [Author Response · Author response to Decision Letter 0]

2 Aug 2023

We thank the anonymous reviewers for reading the manuscript and for his/her effort in providing constructive criticism and positive feedback. A detailed reply to each reviewer's comments is provided below. We have addressed all comments in the revised manuscript. We have also addressed the journal's requirements in the Response to Reviewers Letter.

---

## [Decision Letter · Decision Letter 1]

22 Aug 2023

Comprehensive Tool for a Phase Compensation Reconstruction Method in Digital Holographic Microscopy Operating in Non-Telecentric Regime

PONE-D-23-14581R1

Dear Dr. DOBLAS,

We’re pleased to inform you that your manuscript has been judged scientifically suitable for publication and will be formally accepted for publication once it meets all outstanding technical requirements.

Kind regards,

Ireneusz Grulkowski, PhD

Academic Editor

PLOS ONE

Additional Editor Comments (optional):

Reviewers' comments:

Reviewer's Responses to Questions

**Comments to the Author**

1. If the authors have adequately addressed your comments raised in a previous round of review and you feel that this manuscript is now acceptable for publication, you may indicate that here to bypass the “Comments to the Author” section, enter your conflict of interest statement in the “Confidential to Editor” section, and submit your "Accept" recommendation.

Reviewer #2: All comments have been addressed

2. Is the manuscript technically sound, and do the data support the conclusions?

Reviewer #2: Yes

3. Has the statistical analysis been performed appropriately and rigorously? 

Reviewer #2: N/A

4. Have the authors made all data underlying the findings in their manuscript fully available?

Reviewer #2: Yes

5. Is the manuscript presented in an intelligible fashion and written in standard English?

Reviewer #2: Yes

6. Review Comments to the Author

Reviewer #2: (No Response)

7. PLOS authors have the option to publish the peer review history of their article (what does this mean?). If published, this will include your full peer review and any attached files.

Reviewer #2: No

---

## [Editor Report · Acceptance letter]

29 Aug 2023

PONE-D-23-14581R1 

Comprehensive tool for a phase compensation reconstruction method in digital holographic microscopy operating in non-telecentric regime 

Dear Dr. Doblas:

I'm pleased to inform you that your manuscript has been deemed suitable for publication in PLOS ONE. Congratulations! Your manuscript is now with our production department. 

Kind regards, 

on behalf of

Dr. Ireneusz Grulkowski 

Academic Editor

PLOS ONE